# Effect of Structure on the Thermal-Mechanical Performance of Fully Ceramic Microencapsulated Fuel

**Yi Zhou, Zhong Xiao \*, Shichao Liu \*, Ping Chen, Hua Pang, Yong Xin, Yongjun Jiao, Shixin Gao, Kun Zhang, Wenjie Li and Junchong Yu**

Nuclear Power Institute of China, Science and Technology on Reactor System Design Technology Laboratory, Chengdu 610200, China; YZ012@mail.tsinghua.edu.cn (Y.Z.); PingC123@mail.xjtu.edu.cn (P.C.); 2654805380@mail.cqu.edu.cn (H.P.); xyong042306@xsyu.cn (Y.X.); jyjun18@hrbeu.edu.cn (Y.J.); gsx@alumni.sjtu.edu.cn (S.G.); kunz033@mail.tsinghua.edu.cn (K.Z.); wenjl042@mail.tsinghua.edu.cn (W.L.); yujc1234@hit.edu.cn (J.Y.)

\* Correspondence: xzhong031@mail.tsinghua.edu.cn (Z.X.); hit_lsc@163.com (S.L.); Tel.: +86-028-85908485 (Z.X.)

**Abstract:** The effect of non-fuel part size on the thermal-mechanical performance of fully ceramic microencapsulated (FCMTM) Fuel was investigated, and the non-fuel part size was selected according to integrity maintaining of non-fuel part and silicon carbide (SiC) layers. The non-fuel part size can affect the FCMTM temperature and stress distribution greatly by changing the distance between tristructural isotropic (TRISO) particles. The maximum temperature of SiC matrix increased from 1220 K to 1450 K with the non-fuel part size increasing from 100 μm to 500 μm, and the matrix temperature of all the samples was lower than the decomposition point of SiC ceramics. The maximum hoop stress decreased with non-fuel part size, but the inner part exhibiteda crosscurrent trend. The inner part of the SiC matrix lost structure integrity because of the large hoop stress caused by the deformation of TRISO particles, however, the non-fuel parts of FCMTM pellet may maintain their integrity when the non-fuel part size was larger than 300 μm. SiC layers hoop stress increased with non-fuel part size, and the failure probability of SiC layer was lower than $2.2 \times 10^{-4}$ for the FCMTM pellet with small non-fuel part size. The integrity of non-fuel and SiC layers can be maintained for the FCMTM pellet with the non-fuel part size from 300 μm to 400 μm.

**Keywords:** FCM Pellet; Structure Optimization; Thermal-mechanical performance; Non-fuel part size

## 1. Introduction

Fully ceramic microencapsulated (FCM) fuel isformed by tristructural isotropic (TRISO) fuel particles embedding in SiC matrix [1]. FCM is both an acronym and trademark of a patented technology and is written as FCM$^{TM}$ in this paper. The TRISO particle was constituted of fuel kernel and four coated layers including the low-density carbon buffer layer, the inner and outer pyroltytic graphite layers which surround the silicon carbide (SiC) micropressure vessel. The function of the coated layers and SiC matrix has been discussed [2,3]: the buffer layer offered the space to accommodate the generated fission production; the inner and outer pyroltytic graphite layers are the protective layers which can decrease the stress of SiC layer and protect the SiC shell from the energetic fission product recoil damage. SiC layer is the main structural layer which can prevent fission gas release. The functional coated layers and dense SiC matrix of FCM$^{TM}$ offered excellent oxidation resistance and fission product capability, high thermal conductivity and irradiation stability [4,5]. FCM$^{TM}$ pellets are contained by zircaloy cladding for using in light water reactors (LWR) and small modular reactors (SMRs) [6,7].

Obvious safety characteristics, including lower operation temperature and radionuclide production release, can be obtained by using FCM$^{TM}$ fuel [3]. However, FCMs with UO$_2$ or UCO kernel possessed lower fissile loading compared with the UO$_2$ pellet, which cannot satisfy loading requirement of reactor core and limit the application of FCM$^{TM}$ fuel [8]. UN kernels are currently under investigation to increase the fissile loading [8,9]. TRISO particle with high uranium dense has been fabricated in order to increase the uranium loading. U-C-N was used as TRISO kernel to avoid the reaction between the kernel and the buffer layer [10,11]. The performance of TRISO particles with UN kernel has been investigated in previous reports [12,13].

Feasibility and fuel cycle cost of the FCM$^{TM}$ pellet used in different kinds of reactors has been investigated, such as light water reactor, high-temperature gas-cooled reactor, and small pressurized water reactor [14,15]. An FCM$^{TM}$ pellet can meet the requirements of different kinds of reactors and increase the safety of the reactors, however, application of FCM$^{TM}$ may increase fuel cycle costs. The report on thermal–mechanical performance simulation of FCM$^{TM}$ pellet was rare because of the complex structure, contacting behavior and material properties. The behavior of TRISO particle has been studied vastly and lots of software such as PARFUM, PASTA and STRESS3 has been developed [16]. The performance of TRISO particle can be used as input parameters for the FCMTMsimulation including internal pressure and size deformation [17,18]. The performance of TRISO-based FCMTMfuel in LWR environment was simulated by Schappel and co-workes using BISON [18]. UN was used as TRISO particle kernel to increase the fissile loading of FCMTM. The properties of UN kernel and coated layers were considered including thermal conductivity, swelling and creep of pyrolytic carbon (PyC), thermal conductivity and swelling of chemical vapor deposition (CVD) and nano-infiltration and transient eutectic (NITE) SiC. The simulated results of TRISO particle which calculated by BISON and PARFUM code were compared. The interaction between particles with different distance was calculated but only half of five particles were reserved in the pellet. FCMTMperformance was calculated by subtracting out of TRISO particles and the maximum temperature and hoop stress of SiC matrix was studied. The performance of coated layers was not detected and the interaction between TRISO particle and matrix was not reflected in the literature. Effect of SiC matrix on the performance of TRISO particle was investigated by Ougouag and co-worker [19]. A two-dimensional model was established by adding SiC matrix on the single TRISO particle, the result indicated that the thickness of SiC matrix can affect the stress distribution of the coated layers and SiC matrix. Stress and temperature distribution of the SiC matrix among different TRISO particles was not discussed. The thermal-mechanical performance and criterion to evaluate the integrity of the FCM$^{TM}$ pellet was discussed in our previous work for the given structure, the effect of FCM$^{TM}$ structure such as TRISO particle distance and non-fuel part size on the thermal-mechanical performance was not studied [16].

In this paper, the effect of structure on the FCM$^{TM}$ performance was studied for the given TRISO particle fraction (40 vol%), and the structure was optimized to decrease stress and failure probability of SiC layer and non-fuel part. The performance of the FCM$^{TM}$ pellet with different non-fuel part size was simulated by using two-dimensional model. The internal pressure and size variation of TRISO particles was calculated by using 1/8 characteristic unit, and the results was used as input parameters for FCM$^{TM}$ simulation. The effect of FCM$^{TM}$ structure on the maximum temperature and hoop stress of SiC matrix was studied.

## 2. Materials

Five different kinds of materials were used in FCMTM pellet in this paper, including Uranium Nitride (UN), porous carbon buffer (buffer layer), dense pyrolytic carbon (PyC layer), chemical vapor deposition SiC (SiC layer), and sintered SiC matrix material (SiC matrix). The properties of the coated layers and SiC matrix have great effect on the performance of the FCMTM pellet. UN was chosen as kernel to increase the uranium loading of reactor core. UN has excellent compatibility with buffer layer and no reaction was occurred between kernel and carbon, which can decrease the internal pressure obviously [20]. The buffer layer offered the space for accommodating the fission production because of

the high porosity of buffer layer. The UN kernel and buffer layer weresubtracted out when simulate the FCM^TM pellet behavior because the gap appeared, and no interaction occurred between the buffer layer and the other coated layers [17,18]. Inner and outer PyC layers are designed to provide mechanically pliable layers supporting the SiC micro-pressure vessel [17]. SiC matrix and layer in TRISO particles are expected to offer improved containment of fission production under accident condition. At the same time, SiC matrix possessed higher thermal conductivity compared with $UO_2$, which can decrease the operation temperature obviously for comparative rod average linear power [18].

### 2.1. Uranium Nitride

Thermal conductivity of UN ($K_{UN}$) is the function of burn-up ($Bu$), temperature ($T$) and porosity ($p$) [21]. The thermal expansion coefficient ($\alpha_{UN}$) and specific heat ($C_{p,UN}$) are the functions of temperature ($T$). The relative properties can be expressed by the following equations [21]:

$$K_{UN} = 1.37T0.361\frac{1-P}{1+P}(1-0.025Bu) \tag{1}$$

$$\alpha_{UN} = 7.096 \times 10^{-6} + 1.409 \times 10^{-9}T \tag{2}$$

$$C_{p,UN} = 205.3834 \cdot \left(\frac{365.7}{T}\right)^2 \cdot \frac{\exp\left(\frac{365.7}{T}\right)}{\left(\exp\left(\frac{365.7}{T}\right)-1\right)^2} + 0.0381T + 1.061 \times 10^{12} \cdot T^{-12} \exp\left(-\frac{18081}{T}\right) \tag{3}$$

The irradiation swelling of UN kernel was measured in space reactor; the swelling rate of UN kernel had great influence on the dimensional change of coated layers [13]. UN kernel swelling was given by

$$\frac{\Delta V}{V} = 4.768 \times 10^{-11}T^{3.12}Bu^{0.83}\rho^{0.5} \tag{4}$$

where $\Delta V$ and $V$ are the volume increment (%) caused by irradiation and the initial volume respectively. $T$ is the temperature (K), $Bu$ is the burnup (MW·d/TU), and $\rho$ is the density of UN kernel (kg/m$^3$).

Fission gas (Xe, Kr, and He) release was calculated according to two mechanisms: the recoil release and diffusion release. The release of Xe and Kr was controlled by the two mechanisms but the release of He will be only considered coming from diffusion release mechanism [22]. The recoil release will be the predominant mechanism at low temperature and its portion of fission gas can be calculated from the following empirical equation [8]:

$$f = \frac{1}{2}\frac{S}{V}\alpha \tag{5}$$

where $f$ is the portion of Xe and Kr, $S$ is the superficial area of fuel particle (m$^2$), $V$ is the volume of fuel particle (m$^3$), and $\alpha$ is the mean recoil range of fission gas atoms (m). The mean recoil range of Xe and Kr is 3.98 μm and 5.68 μm, respectively [22].

The main mechanism of fission gas release at high temperature is diffusion release, and the Booth classical diffusion model was employed in this study to compute the final release fission gas of loose pyrolytic carbon and the gap through diffusion mechanism. The grain of UN was considered as the ideal sphere of 20 μm in diameter, which means the solving equation can be simplified as 1-dimensional form. The effective diffusion coefficient of fission gas atoms within the fuel grains can be set from the following empirical relation [22]:

$$D_g = 6.66454 \times 10^{-8} \exp\left(-\frac{19164}{T}\right) \tag{6}$$

where $D_g$ is the effective diffusion coefficient of fission gas atoms within the fuel grains.

## 2.2. Buffer Layer

The density of buffer was about 0.9–1.0 g/cm$^3$. Buffer layer is expected to isotropically shrink under irradiation condition, and the gap appeared due to the shrinkage of buffer layer. Internal pressure can be decreased by increasing the thickness of buffer layer. Porosity has great influence on the thermal conductivity of buffer layer. The thermal conductivity of buffer layer can be written as follows [22]:

$$K_{buffer} = \frac{10.9866(1-P)}{1+9P} \tag{7}$$

where $P$ is the porosity of buffer layer (%). The thermal conductivity of buffer layer increased with neutron fluence due to the densification of buffer layer [18].

The elastic modulus of the buffer layer is isotropic, the expression can be written as follows [22]:

$$E_{buffer} = 11.06(1+0.23\Phi) \cdot (0.956+0.00015T) \tag{8}$$

where $\Phi$ is the fast neutron flux ($10^{25}$n·m$^{-2}$) and T is temperature (K).

## 2.3. PyCLayer

High density pyrolytic carbon was used as IPyC and OPyC layers to protect SiC layer from fission production corrosion and matrix defects. The desity of IPyC and OPyC layers are about 1.9–2.1 g/cm$^3$. PyC layers are expected to shrink initially and then the radial direction is expected to expand while the tangential direction is expected to continue shrinking when the neutron fluence reached $2 \times 10^{25}$ n/m$^2$. The radial and tangential irradiation strain of PyC layers are expressed by the following equations [22].

$$\dot{\varepsilon}_r = -0.077 \exp(-\Phi) + 0.031 \tag{9}$$

$$\dot{\varepsilon}_\theta = -0.036 \exp(-2.1\Phi) - 0.01 \tag{10}$$

where $\dot{\varepsilon}_r$ and $\dot{\varepsilon}_\theta$ are the radial and tangential irradiation strain (%), $\Phi$ is the fast neutron flux ($10^{25}$n·m$^{-2}$).

Elastic modulus of the PyC layer is anisotropic and can be calculated by the following equation [22]:

$$E_{PyC} = 25.5(0.384+0.000324\rho_{PyC})(0.481+0.519BAF) \\ (1+0.23\Phi)(0.9560275+0.00015T) \tag{11}$$

where $\Phi$ is the neutron flux ($\times 10^{25}$ n/m$^2$), and T is the temperature (K), BAF is the anisotropic parameters of PyC.

PyC and buffer layers exhibit same creep strain, the expression was given by the following equation [21]:

$$\dot{\varepsilon_{cr,r}} = K_{pyc}[\sigma_r - v_c(\sigma_\theta + \sigma_\Phi)]\dot{\Phi} \tag{12}$$

where $\dot{\Phi}$ is the fast neutron flux rate ($10^{25}$n·m$^{-2}$·s$^{-1}$), $v_c$ is the Poisson ratio, and $K_{pyc}$ is the creep constant (n·m$^{-2}$·s$^{-1}$).The expression was given by

$$K_{pyc} = 2K_0[1+2.38(1.9-\rho_0)] \tag{13}$$

where $\rho_0$ is the initial density (g/cm$^3$). $K_0$ is the function of temperature, and the expression was written as

$$K_0 = 1.996 \times 10^{-29} - 4.415 \times 10^{-32}T + 3.6544 \times 10^{-35}T \tag{14}$$

where $T$ is the temperature (K).

## 2.4. Silicon Carbide

Silicon carbide experiences irradiation-induced swelling at all temperatures, but the swelling mechanism of SiC is different at different temperatures. The point defects swelling may be the dominant mechanism at relative low temperature. At higher temperature (1173–1673 K), Frank faulted loops of interstitial type was the dominant defects, and the Frank faulted loops appear for temperature neutron irradiation at extremely high doses. The irradiation swelling was the function of temperature and neutron flux [23]. The thermal conductivity was modeled as the function of volume swelling and temperature.

The swelling model of the SiC is the function of temperature and neutron flux, which can be written as follows [23]:

$$\dot{S} = k_s \gamma^{-1/3} \exp\left(-\frac{\gamma}{\gamma_{sc}}\right) \tag{15}$$

where $S$ is the swelling rate (s$^{-1}$); $k_s$ is the coefficient of the swelling rate (dpa$^{-2/3}$); $\gamma$ is the neutron flux (dpa); $\gamma_{sc}$ is the characteristic dose for swelling saturation by the negative feedback mechanism (dpa). The swelling of SiC can be obtained from the time integration of the Equation (15).

$$S = S_s\left[1 - \exp\left(-\frac{\gamma}{\gamma_{sc}}\right)\right]^{2/3} \tag{16}$$

where, $S_s$ and $\gamma_{sc}$ are the function of the temperature, and can be expressed as below:

$$S_s(T) = 0.05837 - 1.0089 \times 10^{-4}T + 6.9368 \times 10^{-8}T^2 - 1.8152 \times 10^{-11}T^3 \tag{17}$$

$$\gamma_{sc}(dpa) = -0.4603 + 2.6674 \times 10^{-3}T - 4.3176 \times 10^{-6}T^2 + 2.3803 \times 10^{-9}T^3 \ (18) \tag{18}$$

The thermal conductivity model of SiC layer was expressed [17] as

$$k = \frac{1}{R_0 + R_{irr}} \tag{19}$$

where $R_0$ and $R_{irr}$ are the thermal resistance (K/W) of the prior- and post-irradiated SiC layer, respectively.

The thermal resistance of SiC layer before irradiation is expressed as below [17]:

$$R_0 = \frac{1}{-3.7 \times 10^{-8}T^3 + 1.54 \times 10^{-4}T^2 - 0.214T + 153.1} \tag{20}$$

The thermal resistance induced by the irradiation can be written [17] as

$$R_{irr} = \frac{1}{6.08 \cdot S} \tag{21}$$

where $S$ is the volume swelling (%).

Thermal conductivity of SiC matrix can be expressed by multiplying a parameter on the thermal conductivity expression of SiC layer because the thermal conductivity exhibited similar variation trend and the value was different between SiC matrix and layer [24]. The parameter was defined as 0.75.

The specific heat, elastic modulus and elastic Poisson Ratio of SiC was set as constant, the values are 1200 J·kg$^{-1}$·K$^{-1}$, 450 GPa, and 0.2 respectively [23].

The creep model of SiC can be expressed as follows:

$$\dot{\varepsilon_{creep}} = K_1 \dot{\Phi} \sigma_e \tag{22}$$

where $K_1$ is a temperature-dependent creep coefficient, the value of $K_1$ was $0.4 \times 10^{-31}$ n/(m$^2 \cdot$MPa), $\dot{\Phi}$ is the neutron flux in n/(m$^2 \cdot$s), and $\sigma_e$ is the effective stress.

Failure probability of the SiC layer was calculated according to the strength and stress distribution of FCM$^{TM}$ pellet. The failure probability of SiC layer can be calculated by the following equation [23]:

$$P = 1 - \exp\left[ -\int_V^1 \left(\frac{\sigma_p}{\sigma_0}\right)^m dV \right] \tag{23}$$

where $V$ is the characteristic volume (m$^3$), $m$ is the Weibull modulus of SiC layer, $\sigma_0$ is the characteristic strength (MPa), and $\sigma_p$ is the true stress (MPa). The characteristic strength and the Weibull modulus ofSiC layer were 350MPa and five, respectively [17].

## 3. Geometry Parameters and Modeling Approach

The kernel diameter was 800 µm which was larger than the traditional TRISO particle (about 450 µm) to increase the fissile loading in order to meet the requirement of LWR. The thickness of the coated layers was designed, the thickness of buffer, IPyC, SiC, and OPyC layers were 100 µm, 30 µm, 40 µm, and 30 µm, respectively.

A FCM$^{TM}$ pellet with 40vol% TRISO particle loading was simulated. FCM$^{TM}$ pellets with different non-fuel part size were design, and the effect non-fuel part size on the performance of FCM$^{TM}$ was investigated. FCM$^{TM}$ pellet with non-fuel part has been fabricated by our group, the microstructure was obtained by using X-ray imaging techniques as shown in Figure 1. The irradiation measurement will be conducted in the future in our own testing reactor. The definition of a non-fuel part isshown in Figure 1. The distance between two TRISO particles decreased with the increasing of non-fuel part size due to the certain TRISO particle loading in FCM$^{TM}$ pellet. The FCM$^{TM}$ pellet samples with different non-fuel part size were labeled. For example, N100 means the non-fuel pare size of the sample was 100 µm.

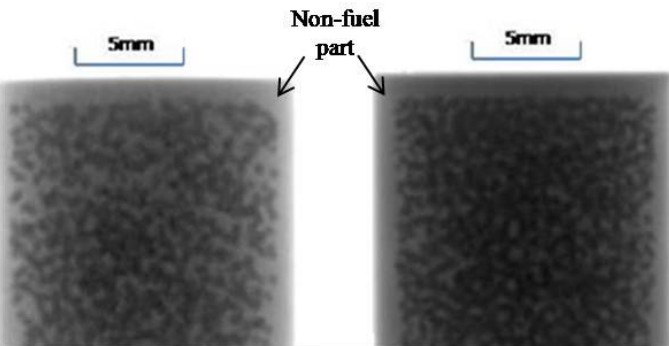

**Figure 1.** Definition of non-fuel part in FCM$^{TM}$ pellet.

The temperature and stress distribution in FCM$^{TM}$ and TRISO fuel was calculated using heat transfer and the solid mechanical model, respectively, which was offered by COMSOL multi-physics software (COMSOL-5.2, MERCURY LEARNING AND INFORMATION LLC, Dulles, Virginia). The heat conduction equation and heat flux used to calculate the FCMTM performance have been introduced in our previous work [17]. The fission gas release was calculated according to two mechanisms including the recoil release and diffusion release. The model of the fission gas release was discussed in above section. The internal pressure was calculated according to the ideal gas low, the temperature, amount of fission gas, and gap volume, which were investigated using the TRISO particle model.

The schematic of the calculation flow used COMSOL-5.2 software, as shown in Figure 2. Here, a 1/8 sphere unit was used to simulate the performance of TRISO particle, the internal pressure under different temperatures were calculated as input parameters for FCM$^{TM}$ simulation. Two-dimensional

models of FCM$^{TM}$ pellet with different structure parameters were established. Solid mechanics and heat transfer modules were selected for normal condition. Materials property models were input by defining the analytic functions. The boundary condition of TRISO particle and FCM$^{TM}$ pellet was defined, symmetry boundary condition and surface temperature were defined for the three sides and surface of TRISO particle respectively. The boundary conditions of FCM$^{TM}$ pellet have been introduced in our previous work [17]. The temperature field, stress distribution and failure probability were output and the structure of FCM$^{TM}$ pellet was optimized.

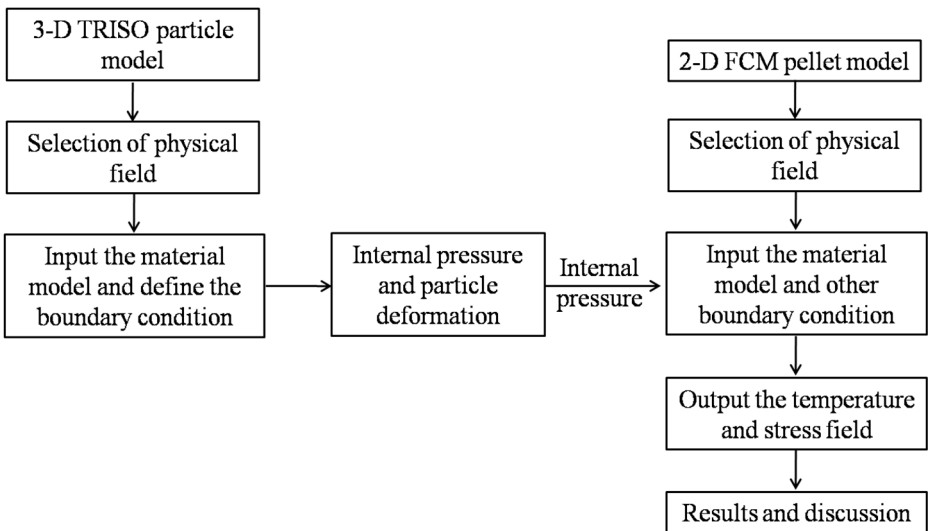

**Figure 2.** Computable flow graph of simulation process for TRISO particle and FCM$^{TM}$ pellet.

## 4. Results and Discussion

### 4.1. Deformation and Fission Gas Release

The dimensional change of TRISO particle can affect the stress distribution of SiC matrix and SiC layer because of the interaction between TRISO particle and SiC matrix. The dimensional change of kernel and coated layers was shown in Figure 3. The deformation of TRISO particle may be caused by the thermal and irradiation deformation of the kernel and coated layers. UN kernel swells as a function of burn-up and temperature shown in equation (4), and the radius of UN kernel increased with operation time. Buffer and IPyC layer deboned and the gap appeared at beginning. This phenomenon was caused by the shrinkage of buffer and IPyC layers. Buffer and IPyC layers shrink because of the irradiation densification and irradiation shrinkage as mentioned in Sections 2.2 and 2.3. The gap size increased firstly because the radius of buffer layer decreased rapidly. Then the gap size decreased due to the swelling of kernel and IPyC layer, this result was good agreement with literature [20]. The radius of OPyC layer decreased slightly because of the irradiation shrink of OPyC layer, the deformation of OPyC provide protection of SiC layer by pressing the inner layer, but the deformation of TRISO particle may induce tensile stress on SiC matrix especially the part among TRISO particles. The radius of IPyC layer decreased with operation time due to the irradiation shrink.

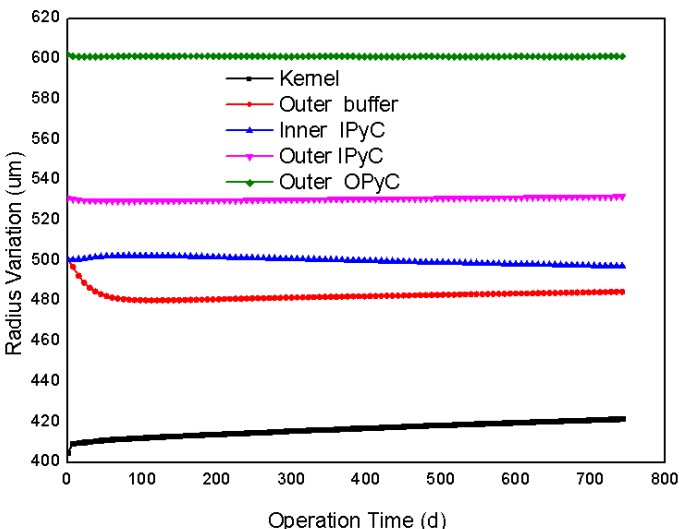

**Figure 3.** Radius variation of kernel and coated layers.

The fission gas release and internal pressure of TRISO particle were shown in Figure 4. The amount of fission gas release including He, Kr and Xe were calculated. The release rate was controlled by recoil and diffusion mechanisms as mentioned above. The release amount of He, Kr and Xe were $4.41 \times 10^{-8}$ mol, $5.85 \times 10^{-9}$ mol and $4.33 \times 10^{-9}$ mol respectively at the end of life. The total amount of fission gas was about $5.43 \times 10^{-8}$ mol. The amount fission gas release increased with operation time linearly, this trend was similar with literature [20]. The internal pressure was approximately linear increased with operation time, which caused by the accumulation of fission gas release [20]. The internal pressure of UN kernel TRISO particle was much lower than the $UO_2$ kernel, because there was no CO or other reactivity gas produced by the reaction between $UO_2$ kernel and buffer layers [12]. The internal pressure of TRISO particle with UN kernel was about 2.2 MPa at the end of life which was much lower than the $UO_2$ kernel (about 45MPa) reported in the literature [25]. UN kernel can decrease the internal pressure obviously by avoiding the reaction between kernel and coated layers.

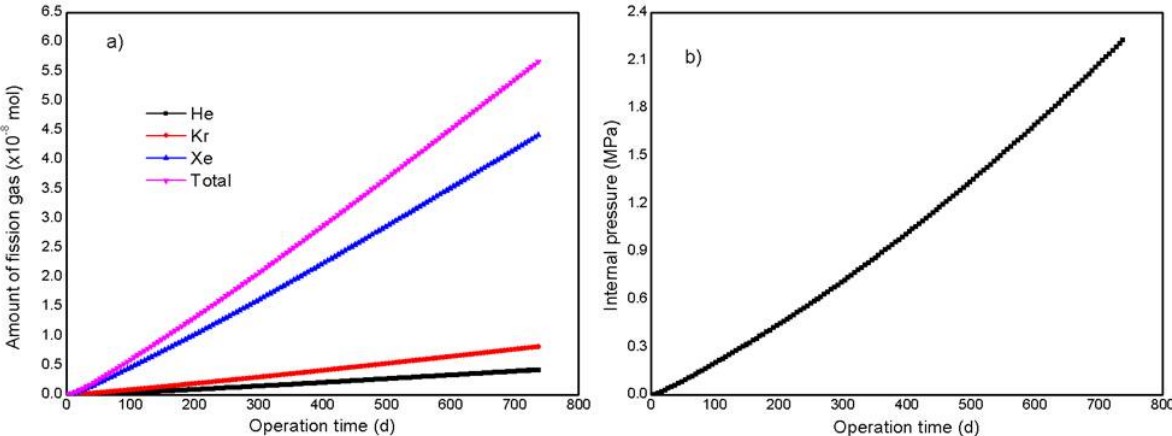

**Figure 4.** Variation of amount of gas release and internal pressure of TRISO particle, (**a**) fission gas release, (**b**) internal pressure.

*4.2. Matrix Temperature*

Temperature variation and distribution of the FCM$^{TM}$ pellets with different non-fuel part size in an LWR environment was shown in Figure 5. The maximum temperature was located in the inner part of FCM$^{TM}$ pellet. The maximum temperature of SiC matrix increased with non-fuel part size, which increased from about 1349K to 1586K. TRISO particles flock together in the sample with large non-fuel

part and the heat transfer path was less compared with the small non-fuel part ones. All the SiC matrix maximum temperature was much lower than the decomposition or molten point of SiC ceramic, which was met the requirements of temperature-limited criteria [24]. The maximum temperature of the samples was lower than $UO_2$ pellet under same condition due to the high thermal conductivity of the SiC matrix. All samples exhibit similar temperature variation trend, maximum temperature of the SiC matrix increased rapidly and then was followed by a slower linear increase. This result was caused by the thermal conductivity variation of SiC matrix. Thermal conductivity of the SiC matrix decreased rapidly at first because of the increasing vacancy accumulation and then saturated [24], and the thermal conductivity of SiC matrix was stable according to Equations (19) and (21).

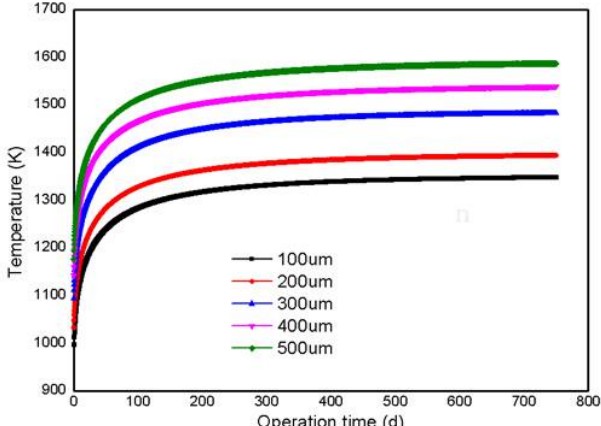

**Figure 5.** Matrix temperature profile at EOL of the FCM$^{TM}$ pellet and variation of maximum matrix temperature with operation time.

### 4.3. Stress Distribution of SiCMatrix

Variation of hoop stress on non-fuel parts of the FCM$^{TM}$ pellets with different non-fuel parts size located in different parts were shown in Figure 6. The maximum hoop stress on non-fuel part decreased with the increasing of non-fuel part size because of the stress concentration on the non-fuel part with small size. Hoop stress on different samples exhibit similar variation trend. The hoop stress decreased firstly and then increased [17]. The SiC matrix swelled rapidly at first, and the OPyC layers shrank. The interaction between SiC matrix and TRISO particle was small. SiC matrix swelling reached saturation at about 1 dpa, the temperature gradient increased and non-uniformity deformation of SiC matrix occurred [19]. The interaction between matrix and TRISO particle increased, which caused the non-fuel part hoop stress increased. The difference between different parts in the same sample was caused by the temperature distribution and interaction among TRISO particles. The maximum hoop stress on non-fuel part decreased from about 1200 MPa to 400 MPa with the non-fuel part size increasing from 100 μm to 500 μm. The non-fuel part hoop stress of the FCM$^{TM}$ pellet with 500 μm non-fuel part size was about 400 MPa, which was similar to the SiC matrix strength. The decreasing of non-fuel part hoop stress was benefit to the integrity maintaining of FCM$^{TM}$ pellet.

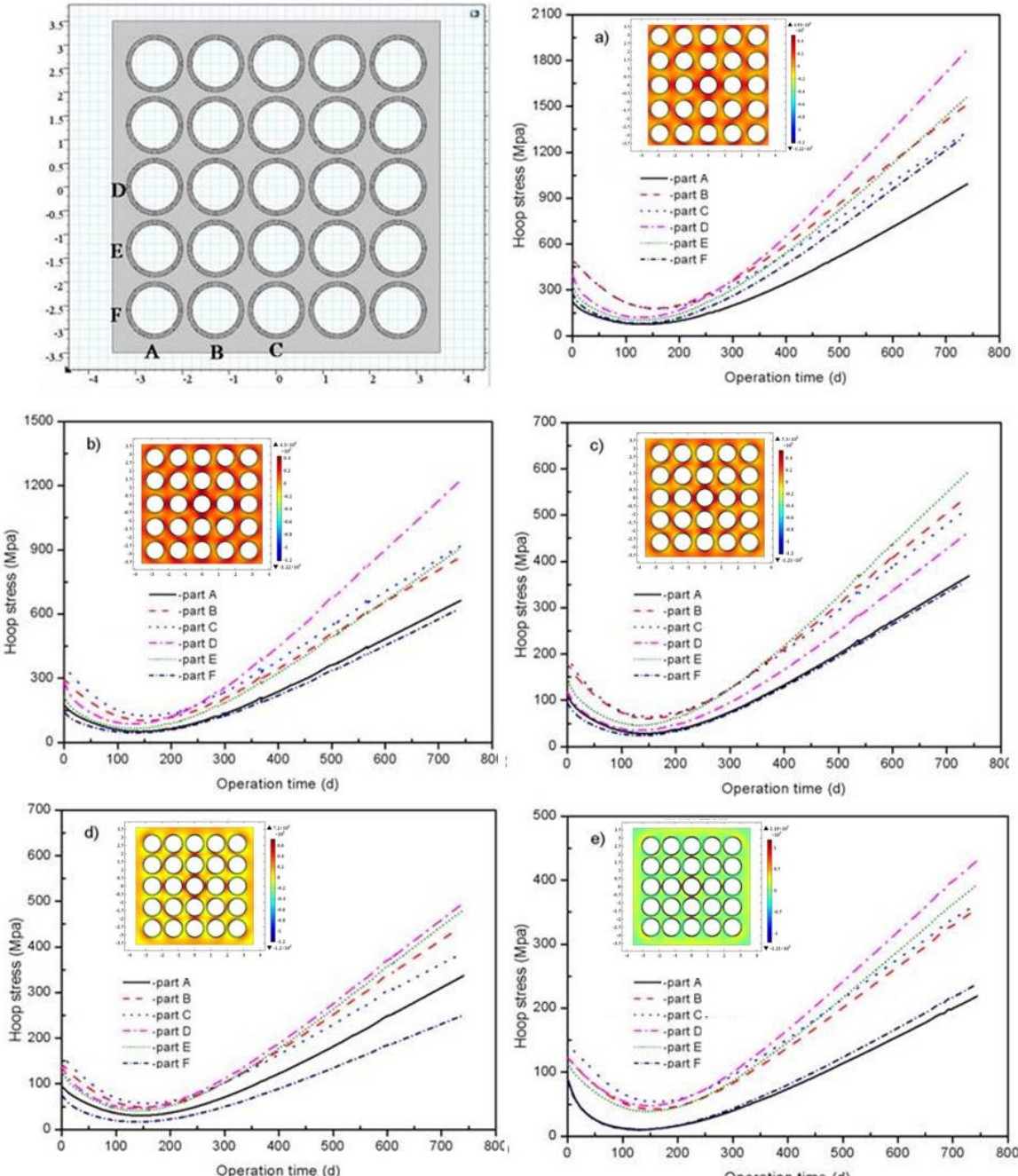

**Figure 6.** Variation and distribution of hoop stress of the non-fuel part in FCM$^{\text{TM}}$ pallet with different non-fuel part size, (**a**) 100 μm, (**b**) 200 μm, (**c**) 300 μm, (**d**) 400 μm and (**e**) 500 μm.

Figure 7 shows the hoop stress of inner SiC matrix located between the two TRISO particles in the FCM$^{\text{TM}}$ pellet with different non-fuel part size. The maximum hoop stress of the inner SiC matrix increased with the non-fuel part size. The maximum inner SiC matrix hoop stress of the pellet with 100 μm non-fuel part was about 900 MPa at the end of life but the value with 500 μm non-fuel part reached 3000 MPa. The increasing of non-fuel part may cause the decrease of distance between TRISO particles. Shearing action between the TRISO particle and the SiC matrix was enlarged because the two TRISO particle next together [16,24]. The hoop stress variation of the inner SiC matrix and non-fuel part are similar, the hoop stress decreased firstly and then increased, which may be caused by the deformation of SiC matrix and TRISO particle. The hoop stress of all samples was much higher than the strength of the sintered SiC ceramics at the end of life. The high hoop stress may have caused the

broken of SiC matrix according to the stress criterion. This result has been proved by Schappel in adoctoral thesis [24].

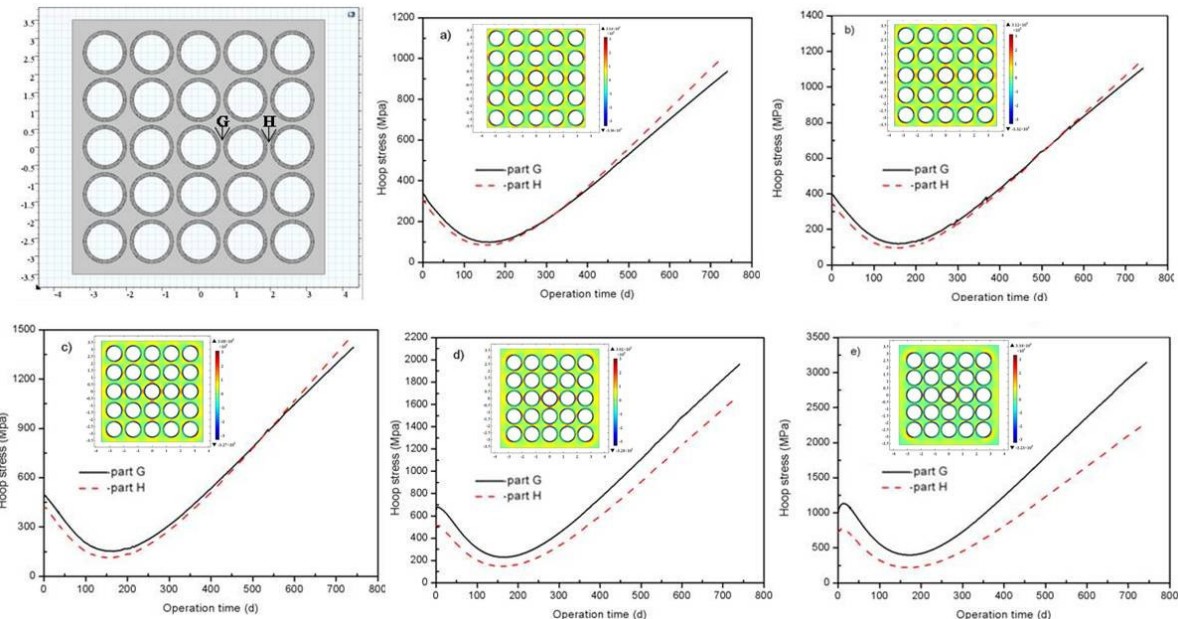

**Figure 7.** Variation of hoop stress of the inner SiC matrix, (**a**) 100 μm, (**b**) 200 μm, (**c**) 300 μm, (**d**) 400 μm and (**e**) 500 μm.

## 4.4. Performance of the SiC Layer

FCM$^{TM}$ pellet failure can be reflected by the integrity of inner SiC matrix, non-fuel part and SiC layers in TRISO particle as we mentioned in the first section [17,18]. SiC ceramics possessed excellent fission products capabilities because of the high density and low transfer diffusion of fission production. Inner SiC matrix was broken due to the high hoop occurred during operation. The hoop stress of non-fuel part was decreased with the increasing of non-fuel part size, and the integrity of non-fuel may be maintained for the FCM$^{TM}$ pellet with large non-fuel part size (>300μm). The performance of SiC layer is an important factor for the evaluation criterion of FCM$^{TM}$ pellet failure and the performance of SiC layers in FCM$^{TM}$ pellet with different non-fuel part size was studied in this work.

The hoop stress of SiC layers embed in different location of FCM$^{TM}$ pellet was shown in Figure 8, and the effect of non-fuel part size on the performance of SiC layers was also discussed. The maximum hoop stress increased with non-fuel part size. The maximum SiC layer hoop stress of the pellet with 100 μm non-fuel part was about 120 MPa at the end of life but the value with 500 μm non-fuel part reached about 400 MPa. Because of the distance between TRISO particle decreased with the increasing of non-fuel part size, the interaction between different TRISO particles enlarged and the hoop stress of SiC layer increased. Maximum hoop stresses for the pellets with 300μm and 400μm non-fuel part size were 120MPa and 200 MPa respectively, which were much lower than the strength of SiC layer [26]. The failure probability of SiC layer can be calculated using Weibull distribution as shown in equation (23). The failure probability of SiC ceramics using Weibull distribution has been reported in previous works [26]. SiC layer located in different parts suffered different deformation, the variation and hoop stress were different for SiC layers located in different pellet parts; this result had been introduced in our previous work [17].

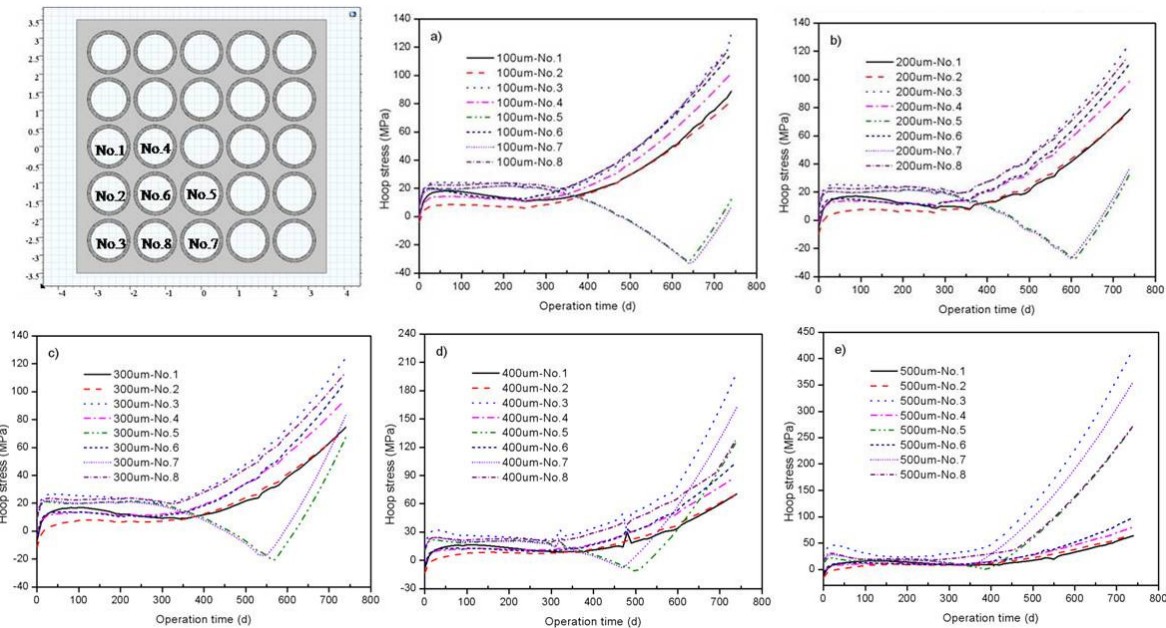

**Figure 8.** Variation of hoop stress on SiC layer located in different parts in FCM$^{TM}$ pellet, the non-fuel part size of pellet was: (**a**) 100 μm, (**b**) 200 μm, (**c**) 300 μm, (**d**) 400 μm and (**e**) 500 μm.

The failure probability of SiC layers in FCM$^{TM}$ pellets with different non-fuel part size was shown in Figure 9. The maximum failure probability of SiC layers increased with non-fuel part size because of the increasing of SiC layers hoop stress. The hoop stress of SiC may be caused by the interaction between TRISO particle and SiC matrix, the contribution of internal pressure and fission gas release was limited [19]. The maximum failure probability of SiC layers in FCM$^{TM}$ pellets with 100 μm and 500 μm non-fuel part was about $9.0 \times 10^{-5}$ and $1.2 \times 10^{-2}$ respectively. Maximum failure probability for the pellets with 300μm and 400 μm non-fuel part size were $1.05 \times 10^{-4}$ and $2.2 \times 10^{-4}$, respectively, so the low failure probability ensured the integrity of the SiC layers when the non-fuel size lower than 500 μm [20].

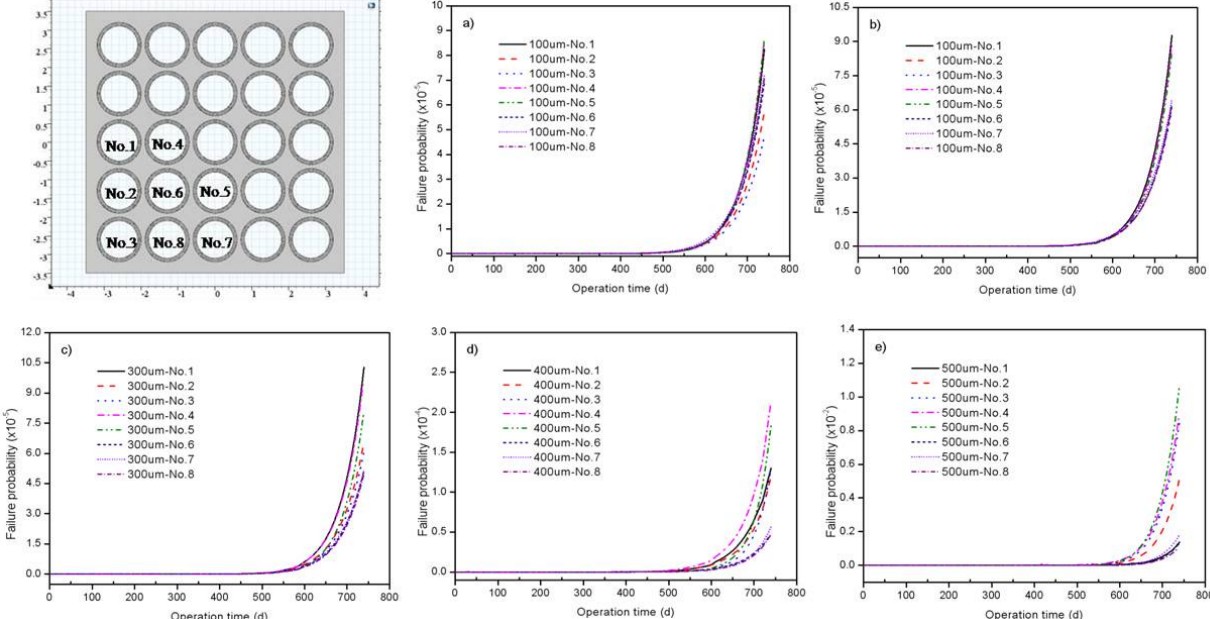

**Figure 9.** Failure probability of SiC layers located in different parts of FCM$^{TM}$ pellet, the non-fuel part size of pellet was: (**a**) 100 μm, (**b**) 200 μm, (**c**) 300 μm, (**d**) 400 μm and (**e**) 500 μm.

## 5. Conclusions

A two-dimensional model of FCM^TM pellet with different non-fuel part size was established to simulate the thermal-mechanical performance of the FCM^TM pellet. The effect of structure on the performance of the FCM^TM pellet was discussed, and structure parameters was selected according to the integrity maintaining of non-fuel and SiC layers. The internal pressure of UN-TRISO particle increased linearly with operation time, and the maximum internal pressure was about 2.1 MPa, which was much lower than the $UO_2$ kernel. The maximum SiC matrix temperature increased with non-fuel part size, and the maximum temperature was about 1586 K. The non-fuel part hoop stress decreased with non-fuel size while the hoop stress of inner matrix exhibited crosscurrent trend. The SiC layers hoop stress and failure probability increased with the non-fuel part. The structure integrity for the non-fuel part and SiC layers may be maintained for the pellet with 300μm and 400μm non-fuel part size. 300μm and 400μm non-fuel part size may be a suitable choice for the FCM^TM pellet in order to maintain the integrity of the FCM^TM pellet and the SiC layers.

**Author Contributions:** Conceptualization, Z.X., S.G. and J.Y.; methodology, S.L.; software, P.C. and W.L.; validation, H.P., Z.X. and K.Z.; formal analysis, S.G. and Y.J.; investigation, J.L.; resources, Y.Z.; data curation, H.P.; writing—original draft preparation, S.L.; writing—review and editing, Y.X.; visualization, Y.X.; supervision, P.C.; project administration, Y.J.; funding acquisition, Y.J. All authors have read and agreed to the published version of the manuscript.

**Funding:** This research was funded by National Natural Science Foundation of China, Grant No. U1867219. Check carefully that the details given are accurate and use the standard spelling of funding agency names at https://search.crossref.org/funding, any errors may affect your future funding.

**Acknowledgments:** Thanks for the support given by Renjie Ran and Xi Qiu on original draft preparation, review and editing.

**Conflicts of Interest:** The authors declare no conflict of interest.

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
