# Peer review of "Effect of Structure on the Thermal-Mechanical Performance of Fully Ceramic Microencapsulated Fuel"

_computation, doi:10.3390/computation8010013_

Round 1

Reviewer 1 Report

Authors;

I have found it easiest to directly edit.  I do not believe I have altered any content.

Blue are suggested changes.

Author Response

  • I have found it easiest to directly edit. I do not believe I have altered any content.Blue are suggested changes.

√ Thanks for your comments and we have revised our manuscript.

Reviewer 2 Report

The paper is well formulated and the results are inline with previous ones presented by Lance Snead, form Oak Ridge, for instance in NEA/NSC/DOC(2013)9, and points into one of the research needs, regarding accident resistance, and so fuel integrity and stability among other properties. I wish the authors would go a bit further on the conclusions but is fine, a contribution in the direction that is expected to be tested. I hope this is parto of an ongoing research that soon will include experimental data and comparison with it.

Author Response

  • The paper is well formulated and the results are inline with previous ones presented by Lance Snead, form Oak Ridge, for instance in NEA/NSC/DOC(2013)9, and points into one of the research needs, regarding accident resistance, and so fuel integrity and stability among other properties. I wish the authors would go a bit further on the conclusions but is fine, a contribution in the direction that is expected to be tested. I hope this is part of an ongoing research that soon will include experimental data and comparison with it.

√ Thanks for your comments and the result will be compared with experimental data in the future.

Reviewer 3 Report

The manuscript of Zhou et al. titled as ‘Effect of Structure on the Thermal-Mechanical Performance of Fully Ceramic Microencapsulated Fuel’ has already been corrected by the authors two times according to reviewer’s comments. Although the manuscript improved, the current version is still not the final. A final reading of the manuscript is strictly required to improve English and correct numerous typing mistakes, as well as the wrong format of references. I suggest using a spell checker. See examples below.

--- Chapter numbering should be consequent. Put full stop, like 1. Introduction, 2. Materials (lines 28 and 83)  

--- Line 47: remove ‘and’ after U-C-N

--- Line 48: remove the full stop after layer

--- Line 51: insert ‘as’ after such

--- Line 93: remove one of the two ‘to’s at the end of the line

--- Equation 1 in line 103 is confluent. Please correct.

--- Line 106: correct ‘measure’ to ‘measured’

--- Line 106: replace ‘,’ with ‘;’

--- Line 107: replace ‘has’ with ‘had’

--- There must be a space between a number and the unit. Please check manuscript thoroughly (e.g. lines 145, 152, 161, 171, 212, 214, 215, 216, 224, 309, 310, 314, 319, 336, 348, 349, 360, 361, 363, 375, and 376)  

--- Line 179: K should be k (see equation 15)

--- Line 191: ‘resistance of before irradiated SiC layer’ should be ‘resistance of SiC layer before irradiation’

--- Equation 23 in line 207 is confluent. Please correct.

--- Line 219: replace ‘showed’ with ‘shown’ ("showed" is the simple past, "shown" is the past participle)

--- Line 220: replace ‘showed’ with ‘shown’

--- Line 234: replace ‘showed’ with ‘shown’

--- Lines 250, 252, 281, 299, 342, 356: replace ‘showed’ with ‘shown’

--- Line 264: replace ‘sowed’ with ‘shown’

--- Line 286: replace ‘was meet’ with ‘met’

--- Line 310: replace ‘similar with’ with ‘similar to’

--- Line 313: Figure 6.

--- Line 330: 4.4.

--- Lines 367-369: The first sentence in the Conclusion part is not clear. Please rewrite (there is a problem with the English sentence structure).

--- Lines 382-442: Provide correct reference styles as required by Computation. Read Instructions for Authors on the journal’s home page.

Author 1, A.B.; Author 2, C.D. Title of the article. Abbreviated Journal Name Year, Volume, page range.

Author Response

  • --- Chapter numbering should be consequent. Put full stop, like 1. Introduction, 2. Materials (lines 28 and 83)  

√ We have revised in our manuscript and labeled with different color.

  • --- Line 47: remove ‘and’ after U-C-N

√ We have revised in our manuscript.

  • --- Line 48: remove the full stop after layer

√ We have revised in our manuscript.

  • --- Line 51: insert ‘as’ after such

√ We have revised in our manuscript.

  • --- Line 93: remove one of the two ‘to’s at the end of the line
  • √We have revised in our manuscript.
  • --- Equation 1 in line 103 is confluent. Please correct.

√ We have revised in our manuscript.

  • --- Line 106: correct ‘measure’ to ‘measured’

√ We have revised in our manuscript.

  • --- Line 106: replace ‘,’ with ‘;’

√ We have revised in our manuscript.

  • --- Line 107: replace ‘has’ with ‘had’

√ We have revised in our manuscript.

  • --- There must be a space between a number and the unit. Please check manuscript thoroughly (e.g. lines 145, 152, 161, 171, 212, 214, 215, 216, 224, 309, 310, 314, 319, 336, 348, 349, 360, 361, 363, 375, and 376)  

√ We have revised in our manuscript.

  • --- Line 179:K should be k (see equation 15)

√ We have revised in our manuscript.

  • --- Line 191: ‘resistance of before irradiated SiC layer’ should be ‘resistance of SiC layer before irradiation’

√ We have revised in our manuscript.

  • --- Equation 23 in line 207 is confluent. Please correct.

√ We have revised in our manuscript.

  • --- Line 219: replace ‘showed’ with ‘shown’ ("showed" is the simple past, "shown" is the past participle)

√ We have revised in our manuscript.

  • --- Line 220: replace ‘showed’ with ‘shown’

√ We have revised in our manuscript.

  • --- Line 234: replace ‘showed’ with ‘shown’

√ We have revised in our manuscript.

  • --- Lines 250, 252, 281, 299, 342, 356: replace ‘showed’ with ‘shown’

√ We have revised in our manuscript.

  • --- Line 264: replace ‘sowed’ with ‘shown’

√ We have revised in our manuscript.

  • --- Line 286: replace ‘was meet’ with ‘met’

√ We have revised in our manuscript.

  • --- Line 310: replace ‘similar with’ with ‘similar to’

√ We have revised in our manuscript.

  • --- Line 313: Figure 6.

√ We have revised in our manuscript.

  • --- Line 330: 4.4.

√ We have revised in our manuscript.

  • --- Lines 367-369: The first sentence in the Conclusion part is not clear. Please rewrite (there is a problem with the English sentence structure).

√ We have rewrote this sentence in our manuscript as “Two-dimensional model of FCMTM pellet with different non-fuel part size was established to simulate the thermal-mechanical performance of the FCMTM pellet. The effect of structure on the performance of FCMTM pellet was discussed and structure parameters was selected according to the integrity maintaining of non-fuel and SiC layers.”

  • --- Lines 382-442: Provide correct reference styles as required by Computation. ReadInstructions for Authors on the journal’s home page. Author 1, A.B.; Author 2, C.D. Title of the article. Abbreviated Journal Name Year, Volume, page range.

√ We have revised in our manuscript.

This manuscript is a resubmission of an earlier submission. The following is a list of the peer review reports and author responses from that submission.

Round 1

Reviewer 1 Report

Typo in Title. The name of the fuel is Fully Ceramic

Intro. When defining FCM, which is a Trademark (FCMTM), the authors should credit the first paper on the subject

L. Snead, K. A. Terrani, F. Venneri, Y. Kim, J. E. Tulenko, C. W. Forsberg, P. F. Peterson, E. J. Lahoda, “Fully Ceramic Microencapsulated Fuels: A Transformational technology for Present and Next Generation Reactors-Properties and Fabrication of FCM Fuel” Transactions of the American Nuclear Society, 104 (2011) 668.

And the patent on the subject

Patent US 9,299,464 B2 March 29, 2016 : Fully Ceramic Nuclear Fuel and Related Methods. F. Venneri, Y. Katoh, and L. L. Snead.

Line 31 is partially true. A BISO, which contains only PyC, also contains significant FP.

Line 35, LWR is light, not lighter. Also, SMR’s do not have to be LWR’s.

Line 36 is not obvious and can be referenced.

Line 38 requires reference.

Line 56. Specific reference for Scappel work

Rewrite sentence starting line 60. It is confusing.

Reference 8 uses Chinthaka, which is the first name of Chinthaka Silva.

An additional, better reference, is Silva, JNM 454 (2014) 405

“non-fuel part” is non-specific.

Line 88 “chosen”

Typo line 92. Also, the densification is not the reason for the FP retention. It is the very low density.

I don’t understand the sentence starting in line 92. I note again that work to date assumes the FP retention is within the inherent porosity in the low density graphite buffer and not any evolving crack, though this will certainly provide additional void space.

Line 95. On purpose of the IPyC is to keep CVD reactant gasses from attacking kernel. The OPyC does not have that function. The main function of IPyC and OPyC is to provide mechanically pliable layers supporting the SiC micropressure vessel.

Line 97 is an overly simplistic statement. I note that FP’s can be released for TRISO without SiC shell rupture and that the SiC matrix was “invented” as an irradiation stable and oxidation resistant replacement for graphite.

Line 174 is an incorrect statement. SiC swells at all temperatures, with the possible exception of a narrow window around 1000C. Below 1000C it is saturation swelling, above 1000C it is void swelling. You should reference the Handbook on SiC properties for Fuels. Snead JNM 371 (2007) 329

Line 202 Typo

Need to continuously define where the equations are derived from lines 188-197

Please define where Fig 1 comes from. Is this original or that of another group’s?

Figure 5 is mis-labeled as the first Figure 6

Line 294. It is incorrect to say that the thermal conductivity has saturated or is in fact in any way due to swelling. Swelling is due to interstitial accumulation. Thermal resistance is due to vacancy accumulation. In the regime you are discussing these are proportional so your statement appears to be correct, but the correct statement is that the thermal conductivity rapidly increases with increasing vacancy accumulation and then saturates. This is discussed in the Handbook reference, along with it’s relation to swelling.

Section 4.2. It is not clear to me which thermal conductivity you are using for the matrix. I believe you are using that of CVD SiC, which is not correct. The NITE matrix is slightly below that of CVD, as discussed in you reference 3

Where is Fig 3 called out?

Your second fig 6 is called out in text at Fig 7.

Figure 9 is after figure 8.

Please figure figure issues…

327 typo TIRSO twice

Have you taken irradiation creep into account? If not, you must.

Rewrite 331/332.

Line 358 needs to be a nuanced statement as strength of SiC follows a Weibull distribution

Author Response

We have revised 

Reviewer 2 Report

The title should not mention "optimizing", since the results mainly discussed the structure effect.

Abstract ==> the long name of FCM needs to be mentioned in the first sentence of abstract.

Numbering of chapters and sub-chapters is not in order. Optimization schematic is not clear.

What was the optimization method that the authors used? All figures are in low quality.

How did the author validate the simulation results of TRISO particle?

How did the authors conduct measurement of data required to estimate temperature field, stress distribution and failure probability of FCM pellet?

Fig. 6 is too small.

Line 149 ==> Density Line 270 - 271 ==> Linear increased -> Proportional Discussion in sub-chapter "Stress distribution of SiC matrix" and "Performance of SiC layer" need more references from other studies.

Line 382 ==> non-fuel Conclusion should also strongly indicate the optimum results from the experiments.

Author Response

We have revised the comments and file was attached.

Round 2

Reviewer 1 Report

1) In original review it was pointed out that the FCM is now an official trademark.  It is therefore required to acknowledge this trademark (TM) when the FCM is first discussed or whenever discussed.  This was considered a mandatory edit which was ignored.  While the authors are correct that FCM is and was a acronym, it is now a trademark.

2) In the original review it was suggested that when the FCM is described the original references to the concept and description be described.  This is common usage. 

3) With respect to irradiation swelling "low" and "high" temperature is not precise terminology.

I have looked over the newly uploaded file and find that my previous comments still hold. I have only required a mandatory revision IF the Journal maintains that trademarked materials be studied (the FCM is a trademark) should be acknowledged as such. The basic issue here is whether the Journal takes a position on publication of results which I believe are being conducted on a patented and trademarked fuel belonging to the USNC without attribution or permission.